# From theoretical models to practical deployment: A perspective and case study of opportunities and challenges in AI-driven cardiac auscultation research for low-income settings

**Felix Krones** [1]*, **Benjamin Walker** [2]

**1** Oxford Internet Institute, University of Oxford, Oxford, United Kingdom, **2** Mathematical Institute, University of Oxford, Oxford, United Kingdom

* felix.krones@oii.ox.ac.uk

**Data Availability Statement:** All data and code are publicly available and addressed through a special section at the end of the paper. Training data:

## Abstract

This article includes a literature review and a case study of artificial intelligence (AI) heart murmur detection models to analyse the opportunities and challenges in deploying AI in cardiovascular healthcare in low- or medium-income countries (LMICs). This study has two parallel components: (1) The literature review assesses the capacity of AI to aid in addressing the observed disparity in healthcare between high- and low-income countries. Reasons for the limited deployment of machine learning models are discussed, as well as model generalisation. Moreover, the literature review discusses how emerging human-centred deployment research is a promising avenue for overcoming deployment barriers. (2) A predictive AI screening model is developed and tested in a case study on heart murmur detection in rural Brazil. Our binary Bayesian ResNet model leverages overlapping log mel spectrograms of patient heart sound recordings and integrates demographic data and signal features via XGBoost to optimise performance. This is followed by a discussion of the model's limitations, its robustness, and the obstacles preventing its practical application. The difficulty with which this model, and other state-of-the-art models, generalise to out-of-distribution data is also discussed. By integrating the results of the case study with those of the literature review, the NASSS framework was applied to evaluate the key challenges in deploying AI-supported heart murmur detection in low-income settings. The research accentuates the transformative potential of AI-enabled healthcare, particularly for affordable point-of-care screening systems in low-income settings. It also emphasises the necessity of effective implementation and integration strategies to guarantee the successful deployment of these technologies.

## Author summary

This study explores the potential and limitations of artificial intelligence (AI) in healthcare, focusing on its role in addressing global health inequities. Non-communicable

https://physionet.org/content/circor-heart-sound/1.
0.3/ Additional test data: https://physionet.org/
content/challenge-2016/1.0.0/.

**Funding:** The author(s) received no specific
funding for this work.

**Competing interests:** The authors have declared
that no competing interests exist.

diseases, especially cardiovascular disorders, are a leading global cause of death, exacerbated in low-income settings due to restricted healthcare access. This research has two components: a narrative literature summary that discusses the gap between AI research and real-world applications, and a case study on heart murmur detection in rural Brazil. The case study introduces an AI model tailored for low-income environments, which efficiently analyses heart sound recordings for diagnostic insights. Both parts highlight the challenges of model generalisation to out-of-distribution data. The findings accentuate the capacity of AI to revolutionise point-of-care screening in resource-limited settings. However, they also highlight the critical importance of effective implementation and conscientious design for the successful deployment of these technologies. By leveraging AI, this work contributes to the broader objective of fostering global health equity, while emphasising the need for thoughtful application and integration strategies.

# 1 Introduction

This paper begins with an introduction to cardiovascular diseases and the PhysioNet Challenge 2022, which forms the basis of the case study. This is followed by an overview of related work and author contributions. As part of an extended introduction, Section 2 offers a narrative literature overview of the opportunities and challenges of AI in healthcare, disparities between income settings, and deployment considerations.

## 1.1 Background: Cardiovascular diseases

Non-communicable diseases are the leading cause of mortality globally. Among the 55 million deaths in 2019, 74% were from non-communicable diseases, as opposed to 18% from communicable diseases and 8% from injuries [1]. Cardiovascular diseases (which form a subset of non-communicable diseases) accounted for 17.9 million (or 32%) of global deaths [1, 2]. These figures are more pronounced in low- and middle-income countries, where over three-quarters of these deaths occur due to limited access to early detection and effective treatment measures [1–3].

Cardiovascular diseases comprise various heart and vessel disorders, such as coronary artery disease, valvular heart disease, and congenital heart disease [4]. Although coronary artery disease is more common in developed nations, congenital and valvular heart diseases are more prevalent in developing countries due to limited prenatal screening and healthcare access. Annually, rheumatic heart diseases account for over 68 million cases and approximately 1.4 million deaths, primarily affecting children and young adults [4]. Early identification of these diseases is important as lifestyle changes can prevent a substantial number of cases. However, the lack of robust primary healthcare often leads to late detection and premature deaths.

By 2030, the World Health Organisation (WHO) aims to reduce the probability that people aged between 30 and 69 years will die from non-communicable diseases to 12.3% (from 17.8% in 2019) [1]. The WHO's strategies include risk factor reduction and improved disease detection. To this end, the WHO has set international objectives, such as lowering the incidence of elevated blood pressure and ensuring 80% availability of affordable basic technologies and medicines for cardiovascular diseases [2]. Achieving these goals necessitates significant investments in health systems, especially in low- to medium-income countries. Thus, cost-effective point-of-care technologies are crucial for heart disease screening in these settings.

Encouragingly, initial results indicate a 27% decline in the individual risk for cardiovascular diseases from 2000–2019 [1].

Anomalies in the early stages of heart structure development can lead to congenital heart disease. While most murmurs do not indicate serious disease, the detection of heart murmurs may serve as an indicator of these structural defects. Early-life heart sound signal analysis could act as a rapid, non-invasive screening method for cardiac structural anomalies, facilitating prompt diagnosis and treatment [5]. Cardiac auscultation and phonocardiography analysis offer straightforward methods for diagnosing heart conditions by identifying abnormal sound waves and heart murmurs in heart sound recordings [5]. The initial stages of heart murmur screening can be relatively straightforward with proper guidance [4]. Nurses can be trained to use a stethoscope effectively and to record heart sounds with technological support. However, interpreting these sounds requires professionals with years of experience, who may not always be readily available. In such scenarios, AI-assisted pre-screening could serve as a viable solution which could aid in the referral of patients to specialised treatment facilities.

## 1.2 Case study context: PhysioNet challenge

In the context of the PhysioNet Challenge 2022 [6], this research builds upon a prior competition submission made by the authors [7]. However, several adjustments were made to align outcomes with the necessities of point-of-care devices in resource-constrained environments and to enhance the models.

The objective of the challenge was to "identify the presence, absence, or unclear cases of murmurs and the normal vs. abnormal clinical outcomes from heart sound recordings" [6]. The scoring mechanism employed a weighted accuracy for the three-class murmur categorisation and a cost function for outcomes classification. The three categories of murmur were: present, uncertain, and absent. The cost function for the outcome classification incorporated: (a) an expert capacity factor, which conveyed the costs associated with patient screening (i.e., when classified as abnormal), (b) significant costs, if a patient exhibited abnormal heart sounds but did not receive treatment, and (c) additional costs, patient treatment [6].

This paper aims to broaden research on this topic by (a) comparing the model using out-of-distribution data in a zero-shot fashion and (b) investigating in more detail underdiagnosis issues.

## 1.3 Related work

Recent reviews have revealed that most current approaches in the classification of heart sounds focus on a binary problem: categorising heart sounds as either normal or abnormal. This emphasis largely stems from the scarcity of available heart sound data which might otherwise facilitate more nuanced classifications [8]. While many studies report accuracies exceeding 90% for heart sound classification tasks (cf. Section 5.2), depending on the task and dataset [9], recent reviews highlight the need to establish robust methods. In terms of deployment and robustness, the reviews identify several challenges [8, 9]. First, the complex, non-stationary nature of heart sound signals complicates their extraction and analysis. Second, the introduction of noise and interference during the acquisition process exacerbates these challenges. Third, the reviews indicate that existing algorithms exhibit limited capabilities and inconsistent accuracy rates, suggesting that they are not yet sufficiently robust for practical, clinical applications. Importantly, these reviews stress the necessity for evaluation using standardised databases for more accurate comparisons of algorithmic performance [9].

### 1.4 Contributions

In this study, the deployment challenges of healthcare technologies are evaluated using a narrative literature review and a case study. A predictive AI model is tailored for heart murmur detection, focusing on resource-constrained environments in rural areas of low-income countries. Additionally, the model's real-world limitations and robustness are assessed, and barriers to practical deployment are discussed.

This paper expands upon a previous work [7], which focused on heart murmur classification and received recognition in the 2022 George B. Moody PhysioNet Challenge [6], securing fourth place. The ultimate objective remains the same: to create an open-source algorithm for accurate classification of heart murmurs using heart sound recordings.

In alignment with the overarching research question—*How can AI technologies be effectively deployed to bridge healthcare disparities between high-income and low-income countries, and what are the opportunities and challenges in achieving this goal?*—the novel contributions of this research are as follows:

- Findings from a literature review and the case study are summarised. The review focuses on the identification of challenges and barriers to deploying AI models for pre-screening in low-income settings. The findings are discussed thoroughly, and challenges are assessed using the NASSS framework [10].

- Expanding upon the contributions of Walker et al. [7], the deep learning model is improved by incorporating multimodal data to extend its generalisability. Two-dimensional spectrograms derived from heart sound recordings are used for the classification of heart murmurs. The improved model is compared to other architectures, including baseline Residual Networks (ResNets) without a Bayesian component.

- A multi-site validation of the refined model and a robustness evaluation are performed, and gaps requiring attention (in order for successful real-world deployment to occur) are identified.

## 2 Review of AI deployment in healthcare

### 2.1 Opportunities and challenges presented by AI in healthcare

**Opportunities.** AI offers an opportunity to enhance areas of healthcare such as diagnostics, treatment planning, and overall patient outcomes, particularly as healthcare demand and costs increase [11, 12]. The capabilities of AI have expanded across various healthcare applications in recent years. For instance, image reconstruction and analysis in radiology have substantially improved due to the integration of deep neural networks [13]. AI-aided detection and diagnosis have the potential to assist healthcare professionals by improving efficiency and accuracy [14, 15]. Furthermore, algorithms that identify areas of interest during image screening have proven effective in supporting clinicians, enhancing diagnostics without supplanting human expertise [14].

**Challenges.** However, the incorporation of AI into healthcare is obstructed by numerous obstacles. These include regulatory hurdles, data privacy concerns, data quality issues, ethical considerations, clinical validation, and funding shortfalls [16]. From a technical standpoint, it is important that models are robust, adaptable, and accurately convey their uncertainty [17].

**Way forward.** Beyond developing a more realistic model evaluation and approaches for better generalisation [18], addressing the challenges mentioned above requires education, collaboration among healthcare providers and industry stakeholders, as well as ongoing

evaluation and refinement of AI systems [19]. To foster acceptance among end-users, early integration of users into development is crucial and sufficient training on the correct use of the technology [16, 20]. Beyond performance expectations, it is essential to meet expectations regarding required effort, social impact of the systems (e.g., on communication or decision-making), and other facilitating conditions such as infrastructure and legal frameworks [20].

## 2.2 Generalisability challenges

**Challenges.** Model reliability on out-of-distribution data (which were unseen during training) is a large concern in medical imagining and medical AI research. The data distribution of medical images can change, for example, due to the variations in imaging equipment, the use of different protocols, or changes in the patient populations across locations and time [18]. This phenomenon, known as feature shift, can significantly impact model performance. Recently, special interest has been directed to population shifts and the ability of models to perform well across different patient subgroups, to ensure fairness and address biases that often stem from unbalanced training data [21, 22]. Additionally, label shift, where the distribution of the labels changes across different datasets, poses another challenge. For instance, in heart sound recordings, the definition of what constitutes an abnormal or normal recording could change over time. Furthermore, variability in human annotations used for training further complicates the issue. Different raters may provide inconsistent labels for the same data (this is known as inter-rater variability), and the same rater may provide inconsistent labels at different times (this is known as intra-rater variability).

**Way forward.** Studies have begun to address these issues in more detail, focusing on a model's ability to function accurately despite out-of-distribution shifts. For example, some researchers evaluate model performance across various datasets [23] and others examine specific data changes, such as temporal variations [24]. Furthermore, extensive studies have been performed on diabetic retinopathy in India [25, 26] and their transferability from HICs to LMICs. For instance, an AI model trained to detect diabetic retinopathy on data collected in Singapore has been shown to maintain its effectiveness when evaluated on data collected in Zambia. This demonstrates that a well-developed AI can be a valuable resource even across sites [27]. To evaluate the generalisability of models, a wide set of metrics must be considered [17], especially metrics that are clinically applicable. This involves considering the impact of varying error rates beyond a narrow set of fairness metrics and acknowledging additional factors like absolute welfare or priority. This is important to prevent Pareto inefficient outcomes, a situation in which enhancements in model performance for one group could still be realised without negatively impacting other groups, as described by Mittelstadt et al. [21].

**Gap.** Despite its importance, research indicates that 72% of recent clinical machine learning studies do not include multi-site evaluation [28]. This suggests a considerable gap in the current approach to AI deployment in healthcare. Many AI models in the literature initially appear to outperform human practitioners but failed to maintain their superiority under more variable testing across multiple sites. The gold standard in testing AI models is the use of randomised controlled trials. However, these trials have only been conducted in a few dozen studies [29]. One study, for example, used a randomised trial to investigate performance, costs, and treatment time for HIV-Tuberculosis screening in Malawi [30]. While many studies only involve small cohorts, their increased instance is a positive step forward.

**Contribution.** This study tests a heart murmur model across multiple sites using publicly available data. As illustrated in Table 1, not many databases exist which are comparable in size to the 2022 Challenge data. Available databases do not all contain multimodal data and often only have short and clean recordings available. Given that the PhysioNet/CinC Challenge 2016

**Table 1. Overview of publicly available heart sound databases with more than 500 recordings.**

| Name | Rec. [#] | Freq. [Hz] | Durat. [sec] | Labels | Patients [#] | Location | Demographics |
|------|----------|------------|--------------|--------|--------------|----------|--------------|
| PhysioNet 22 [4] | 5272 | 4000 | 5–80 | Murmur: | 1568 | Available | Available |
| | | | | Present | | | |
| | | | | Absent | | | |
| | | | | Unknown | | | |
| | | | | Outcome: | | | |
| | | | | Normal | | | |
| | | | | Abnormal | | | |
| PhysioNet 16 [31] | 3153 | 2000 | 5–120 | Normal | 764 | Partially | Partially |
| | | | | Abnormal | | | |
| Yaseen [32] | 1000 | 8000 | 1–4 | Normal | na | na | na |
| | | | | Aortic Stenosis | | | |
| | | | | Mitral Stenosis | | | |
| | | | | Mitral Regur. | | | |
| | | | | Mitral Prolapse | | | |
| Pascal B [33] | 656 | 4000 | 1–25 | Normal | na | na | na |
| | | | | Murmur | | | |
| | | | | Extrasystole | | | |

database is the largest available resource with multimodal data, it was selected for our multi-site evaluation. As most databases primarily contain labels for normal/abnormal classification, we focused on this outcome classification during the multi-site assessment.

## 2.3 Healthcare AI in low-income settings

**Motivation.**   Despite considerable strides toward achieving the health-related sustainable development goals set by the WHO [1], a pronounced discrepancy still exists between the health outcomes and available health resources in high-income countries (HICs) and their low- or medium-income countries (LMICs). For instance, in 2020, a global shortfall of 15 million health workers was reported [1], a gap that is notably wider in LMICs than in HICs. The disparity is stark: Europe reported an average of 36.6 medical doctors per 10,000 citizens, whereas there are only 2.9 in Africa and 7.7 in South-East Asia. Such disparities highlight the diverse needs and potential applications of AI technologies across different resource settings. In developed nations, a major use case of AI is the improvement of individualisation and efficiency of healthcare. By contrast, in low-income settings, a major use of AI is to bridge healthcare delivery gaps. For example, while citizens in HICs may have immediate access to medical professionals, a pressing need exists for simplified pre-screening systems in LMICs, which can be administered by frontline healthcare workers. AI can facilitate task shifting, enabling community health workers to deliver more services [34]. Technologies like AI-driven heart sound interpretation can offer initial pre-screening for cardiac conditions in areas where doctors are scarce. Consequently, AI has the potential to significantly enhance both the quality and quantity of healthcare in LMICs [27, 35–37].

**Recent development.**   A myriad of machine learning models for healthcare have been developed recently, many of which are intended to aid LMICs. The typical objectives of many recent technologies for LMICs are either to assist frontline healthcare workers [38] (e.g., with user-friendly screening tools) or to aid non-specialist clinicians (e.g., non-radiologists) in the analysis of X-rays. For instance, recent work by Rajpurkar et al. [39] shows that an AI system

for chest radiograph interpretation, when combined with input from a non-radiology resident, achieved performance metrics comparable to those of board-certified radiologists. Examples of other studies include COVID-19 forecast models in Iran [40] and India [41], Ebola forecast models for Africa [42], and automated malaria diagnostic models in Uganda [43]. Various tuberculosis prediction studies in Brazil [44], South Africa [45], and Peru [46] have also been conducted. And despite the mixed performance of an AI COVID CT diagnosis tool in Ecuador, it remained in use due to the absence of alternatives [47].

## 2.4 Deployment in low-income settings

**Deployment research.** Recent research efforts have begun to assess the deployment of AI technologies in LMICs. Studies by Okolo [48] and Ismail et al. [38] have examined AI usage among frontline healthcare workers in India, highlighting key design considerations for future applications. Another pilot study considered frontline healthcare workers in Mexico, performing tasks such as triaging palpable breast lumps using an AI-based computer-assisted diagnosis tool with a low-cost portable ultrasound system [49]. A study by Kisling et al. [50] considered automated radiation planning in South Africa to reduce maximum dosage in cervical cancer treatment.

**Open challenges.** The deployment of point-of-care (POC) screening technologies (such as heart murmur detection) in low-income settings presents numerous challenges. Reports indicate inconsistent reliability, varied effects on operational processes, a deficiency in user-centred design, and incompatibility with regional particularities as frequent issues [35, 36, 51]. Limited resources pose pronounced challenges in regions where infrastructure is inadequate, including electricity and internet availability, which are both essential for the operation of POC devices [37]. The effective implementation of POC technologies relies heavily on the availability of a well-maintained supply chain, which is often lacking in low-income settings. Moreover, even with a robust supply chain, the logistical challenges of maintaining and updating complex technological systems in these settings can be formidable. For the development, barriers include constraints in data accessibility, demonstrable financial non-viability [37, 52], as well as concerns surrounding the openness of the data and computation methods involved in training AI tools [51].

**Way forward.** An evidence-based approach is crucial for the successful deployment of POC technologies in low-income settings [37]. This includes conducting thorough risk assessments, considering the unique challenges and limitations of each setting, and prioritising sustainable, long-term solutions that can be integrated into existing systems. Ethical considerations, such as the fair and secure use of AI applications, must also be at the forefront of these efforts [16, 51, 53]. Solutions should focus on integrating intelligence into existing systems and institutions rather than attempting to replace them or build from scratch [37]. Training local healthcare workers and technicians to use and maintain the technology can enhance sustainability and acceptance, fostering a sense of ownership and capability within the community [16, 20]. These strategies help to ensure that the deployment of POC technologies is both effective and enduring.

**Human-centred development.** Most recent research tends to emphasise human-centred development. Research shows that the early integration of the end-users can foster a wider acceptance of the technology [20, 48, 54]. This involves engaging with local communities to understand their specific needs and constraints, ensuring that technologies are user-friendly and culturally appropriate. Notable projects such as Google's automated retinal disease assessment in Thailand and India [54, 55] are examples of this. This project, in collaboration with various clinics, included a human-centred observational study to examine the consequences of

the algorithm's implementation on clinical processes, and to identify factors influencing the performance of the system's algorithm. By 2023, it had screened more than 200,000 people, revealing challenges in data quality, workflow integration, and post-deployment monitoring when shifting into the real world.

**Data-centric development.** AI models trained on global data often require local fine-tuning. This places further burdens on limited local resources and raises questions about the inclusivity and fairness of these technologies in regions with limited data availability [16, 53]. Ensuring standardised data collection in low-income settings is challenging but not impossible. With strategic investments and innovative approaches, such as mobile data collection tools and community engagement, it is feasible to improve data quality. Partnerships with local institutions and the use of scalable technologies can also help standardise data collection processes in these regions.

**Conclusion.** A collaborative ecosystem is important for the success of AI applications in health, including a regulatory framework that provides principles and standards for data governance and a sustainable financing. Open source frameworks present an important step to lower barriers [56]. One of the biggest barriers currently is that data collection and storage are too fragmented and inaccessible [35], which are problems that HIC and LMIC share. Evidence has shown that a human-centred approach is important for the success of tools [48, 55]. A development process in which all stakeholders are considered, and ethnographic fieldwork is conducted (which includes front-line healthcare workers, such as community health workers) is important [48].

## 2.5 Commercial examples of healthcare AI in low-income settings

As Okolo [48] has noted, many studies are steered by large tech companies, such as the Google studies [54, 55]. However, an increasing number of smaller companies are currently working to deploy AI technologies.

Several enterprises and organisations are collaborating with researchers to deploy AI technologies in LMICs. These include Wadhwani AI [57] an Indian company developing AI tools to reduce morbidity and mortality among mothers and children), as well as other eHealth, dermatology, and ophthalmology tools. Aidoc [58], based in Israel and the US, is developing AI tools for cardiovascular and neuroscience diseases with a focus on radiology, care coordination, patient management, and clinical trial enrolment. Ubenwa AI [59], a Nigeria and Canada based company, is developing a computer-aided diagnostics tool for perinatal asphyxia using infant cry sounds. Other organisations, such as OpenMRS [60] and DHIS2 [61], provide medical record systems to countries worldwide.

RAD-AID is a non-profit committed to enhancing radiology resources in low-income environments [62] and its developers are working to overcome issues associated with AI implementation for medical imaging in resource-scarce settings. This non-profit has tackled the shortage of equipment, professional expertise, and infrastructure which typically exist, and has defined data-rights policies. Moreover, RAD-AID has directed attention to addressing the trustworthiness of AI underpinned by a lack of data diversity and the opacity of algorithms. RAD-AID has introduced a triad strategy of clinical radiology education, infrastructure development, and staggered AI deployment. The organisation highlights that AI implementation in LMIC necessitates a strategy that is distinct from that in HICs due to variations in resources and clinical scenarios.

More research is necessary to ensure real-life diagnostic accuracy of commercially available tools. Lind et al. [63] provide some insight, analysing the performance of four chest radiography AI tools on 2,040 patients. Their findings indicate that tools are designed to behave

conservatively. While the authors report moderate to high sensitivity, more false-positive findings were indicated than in comparable radiology reports. They also found that there was decreasing performance for smaller targets and for cases with multiple findings. This highlights the opportunities for AI based screening methods but emphasises the necessity of a careful deployment.

## 3 Materials and methods

### 3.1 Training data

The main training data used in this research were collected by Oliveira et al. [4]. The heart sound recordings were gathered using a Littmann 3200 stethoscope (an online search revealed that this stethoscope is priced between £250 and £300 in the UK [64]) and tablet-based GUI software named DigiScope Collector. This software provides a user-friendly interface for collecting patient metadata and offers clear guidance on the process of recording heart sounds. The heart sound recordings of 1,568 individuals were obtained from an initial pool of 2,061 participants (participants were filtered based on eligibility criteria as stated in the study) during two screening campaigns in 2014 and 2015. These campaigns, known as 'Caravana do Coração', took place in the state of Paraíba in northern Brazil. Mobile teams travelled across the state during the campaigns, collecting data predominantly from a paediatric population. Notably, 63% of the participants were children, and 20% were infants [4]. From the original 1,568 patients, 53.2% were referred for a follow-up, while 36.7% were discharged entirely. The remaining 10.1% either needed additional testing (27 patients), were indicated for surgery or intervention (35 patients), or had no information recorded on their cases (97 patients).

The dataset includes heart sound recordings ranging from 5 to 80 seconds in length, along with demographic information such as age groups, gender, height, weight, and pregnancy status. From the 1,568 patients, 60% (942 individuals) of the recordings were provided for training. A patient could have heart sound recordings from up to six different recording locations, with a total of 5,272 recordings in the dataset. The possible locations of the heart sound recordings were the pulmonary valve, aortic valve, mitral valve, tricuspid valve, or an unspecified location. Furthermore, each patient was assigned two tags, one indicating the presence, absence, or uncertainty of heart murmurs, and the other indicating whether the recordings contain normal or abnormal heart sound recordings. About 13% of the data contained missing values in the metadata, which most commonly occurred concurrently in the age, height, and weight categories.

The recordings were methodically sampled by Oliveira et al. [4] using various algorithms to detect and define the primary heart sounds and their respective boundaries. Labels were assigned to sections of the data that cardiac physiologists deemed to be representative, high-quality segments. The remaining data may comprise both low and high-quality data. In their research, Oliveira et al. [4] sampled signals at 4KHz, because oversampling notably beyond the Nyquist limit (double the highest frequency of the intended signal) offers no extra insights about the signal [4]. Moreover, the heart sound signals were normalised within the range [-1, 1].

### 3.2 Model evaluation

For the in-distribution evaluation, a ten-fold cross-validation was used. As evaluating models in a multi-site context is important in ensuring their safe and effective implementation in real-world settings, as discussed in Section 2.2, the PhysioNet 2016 Challenge database [31] was used for an out-of-distribution evaluation. The database's heart sound recordings were procured from numerous contributors worldwide, collected in both clinical and non-clinical

environments from healthy individuals and patients with heart diseases. The challenge's training set includes a total of 3,153 recordings, each lasting between 5 and 120 seconds. The records correspond to different body locations, typically the aortic, pulmonic, tricuspid, and mitral areas. They are categorised as either normal (79%) or abnormal (21%), with abnormal recordings coming from patients with confirmed cardiac diagnoses, including heart valve defects and coronary artery disease. These recordings involve both children and adults, with each subject contributing between one and six heart sound recordings. All recordings were re-sampled to 2,000Hz and are in .wav format. Regrettably, the database does not allow for the linking of multiple locations to a single patient. Subject identifiers are available only for 490 recordings out of 3,153. Hence each recording was treated as an individual patient with one recording. For zero-shot performance the whole dataset was used for evaluation and for the fine-tuned evaluation a split (of 70% and 30%, respectively) was used. To be able to train on a balanced dataset, we sub-sampled the training data.

## 3.3 Data preparation

The data preparation of Walker et al. [7] was built upon and extended. A short-time, windowed Fourier transformation was used to derive the frequency and phase component of segments of a signal as it changes over time [65]. These features were represented with a spectrogram, which is an image depicting the change in amplitude (or power) of various frequency components over time. Owing to its effectiveness in a range of recent audio classification tasks, the spectrograms have a logarithmic mel scale for the frequency. This is intended to preserve the distance between pitches perceived by humans (cf. Fig 1) [6, 66]. The extraction was performed using the SCIPY and LIBROSA Python libraries. The recordings were segmented into overlapping sections using a window of 4 seconds and a stride of 1 second. The spectrogram of each section was computed using a Fast Fourier Transform with a periodic Hanning window of 25 milliseconds, a stride of 10 milliseconds, a minimum frequency of 10Hz, and a maximum frequency of 2000Hz.

The demographic data were processed following the guidelines set by the organisers of the challenge [6]. This processing step included converting age categories to their approximate equivalent in months, applying one-hot encoding to gender data, and transforming pregnancy status into a binary format. Missing data were addressed using mean imputation. The features extracted from the signals encompassed summary characteristics in the time and frequency domains, along with summary measures for spectral centroid, roll-off, and bandwidth.

## 3.4 Models

**3.4.1 Pipeline architecture.** Fig 2 presents a stylised representation of the data and model pipeline. This includes a classification of the individual spectrograms relative to location, aggregation of these classifications across locations, and a multimodal integration of the demographic data and signal features via XGBoost [67]. The classifications of individual spectrograms were aggregated using the arithmetic mean.

**3.4.2 Bayesian neural network.** For the spectrogram classification, two versions of the deep learning model were explored. The first was a standard ResNet50 [68], which has been shown to be very effective in audio-related tasks [69]. This model acted as the baseline. The second model was an approximate Bayesian neural network (BNN) with the same architecture as the baseline model. The second model is referred to as a binary Bayesian ResNet (BBR) model. Both models were initialised using weights obtained from pre-training the ResNet50 on the ImageNet dataset [70].

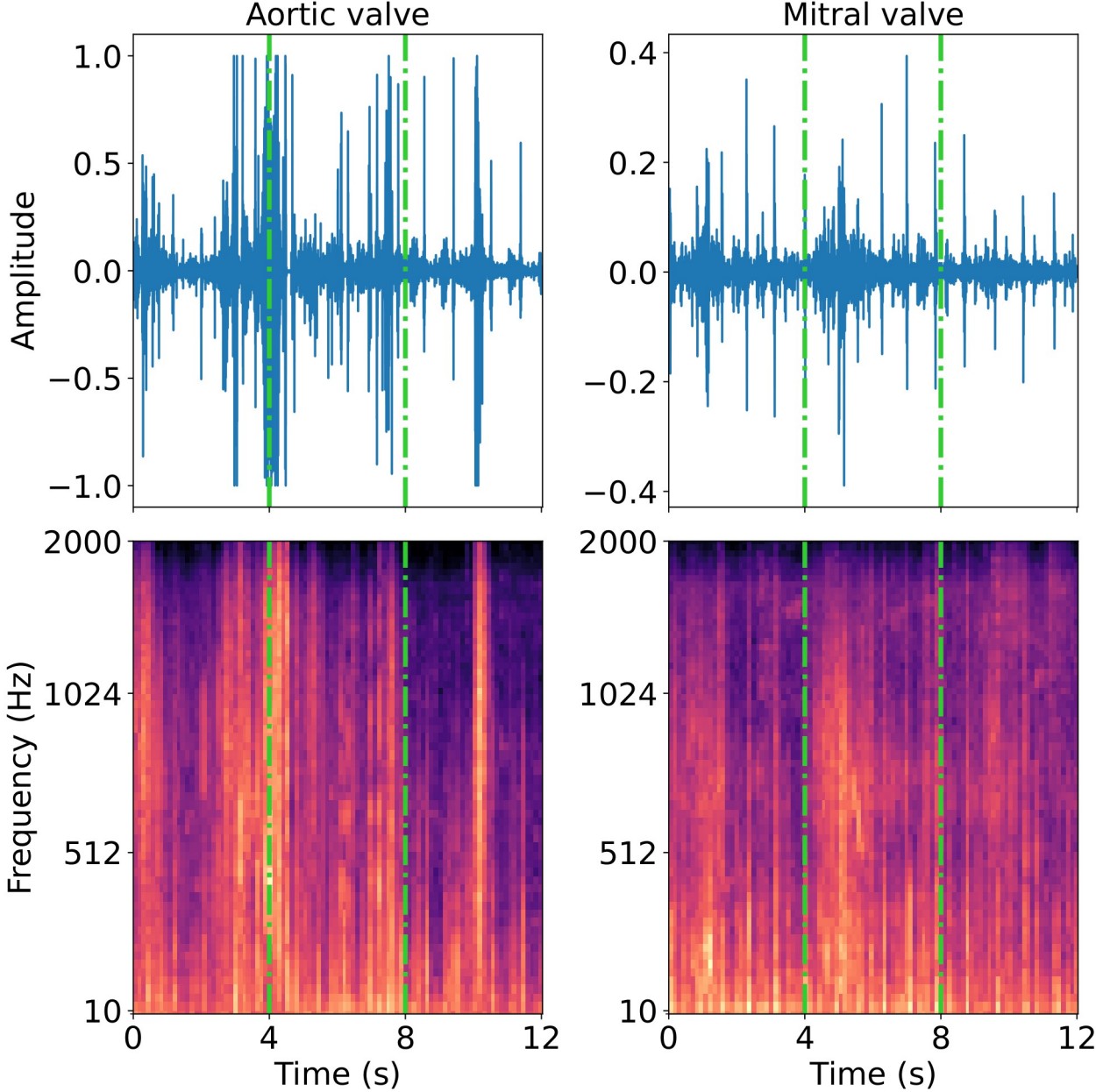

**Fig 1.** "Example heart sound recordings (top row) for a patient with *present* murmur recorded at the aortic valve (left column) and mitral valve (right column). The bottom row shows the log mel spectrogram, as parameterised in the code. The dash-dotted lines show how the data were partitioned into 4 second two-dimensional inputs." From Walker et al. [7].

The parameters of a BNN are distributions instead of fixed values. This means the same input can produce diverse outputs, due to randomness in the model parameters. BNNs are built on the work of stochastic neural networks, which use either stochastic activations or weights to essentially create an ensemble of models. This provides a distribution over outputs and a measure of uncertainty [71]. Research shows that BNNs can reduce overfitting, which is especially beneficial for small datasets like the one examined in the challenge [71]. However, given that this approach does not consistently outperform deterministic counterparts (as

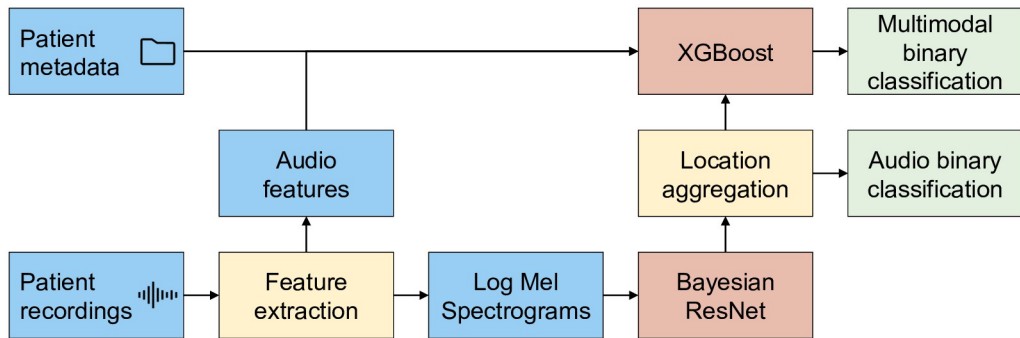

**Fig 2. A schematic diagram of the data and model pipeline.** Key: Blue: Data, Yellow: Fixed methods, Red: Trainable models, Green: Output.

discussed by Kiskin [72]), this research scrutinised the specific impact of this approach within the study.

Constructing a complete BNN requires modelling the prior distribution over all model parameters, a task that can be computationally demanding [73]. However, research has shown that including dropout layers during training and inference is a viable approximation to a complete BNN [71, 73]. Dropout layers are a common component of modern neural networks, which choose a random subset of the neurons to be disabled during each forward pass. Typically, the dropout layers are removed during inference. Inspired by Gal et al. [73], dropout layers were added to various segments of the ResNet50 architecture, particularly to the `BasicBlock()` and `Bottleneck()` modules, as per the ResNet implementation in Kiskin [72]. This can be interpreted as a Monte Carlo approximation to BNNs. The term 'Monte Carlo' signifies the use of random sampling to generate numerical outcomes, specifically creating diverse neural network configurations via the selective deactivation of neurons. (This approach does not strictly approximate BNNs but should still assist in combating overfitting. Further details on the specifics of this approximation are discussed by Gal et al. [73].)

### 3.5 NASSS evaluation framework

To indicate the prospects for scaling up automated heart murmur detection, the NASSS framework was used. The NASSS framework [10] was developed to investigate the challenges associated with the implementation of technologies in healthcare, focusing on the risks of **N**onadoption, **A**bandonment, **S**cale-up, **S**pread, and **S**ustainability. It includes a qualitative guide comprising 19 questions, each of which can be categorised as either simple, complicated, or complex. These questions span seven dimensions: the condition or illness, the technology itself, the value proposition, the system of adopters, the organisational setting, the broader context, and the process of embedding and adaptation over time.

In addition to presenting the findings, this study contextualises them by comparing them with other digital healthcare technologies. Various studies have applied the NASSS framework across different contexts [74]. However, in LMICs the framework has predominantly been used for qualitative evaluation (such as in a study of wearable health monitors in Cambodia [75]), omitting quantitative assessment (simple, complicated, complex). To ensure clarity and avoid ambiguities in the analysis, studies were selected if they included a quantitative dimension. Short of matching studies to ours, the examples include the assessment of telehealth consultations in Australia [76], the adoption of digital twins in healthcare [77], and the

implementation of in-hospital malnutrition screening systems [78], which were all drawn from varied healthcare settings.

## 4 Case study results

### 4.1 Training data analysis

The 2022 Challenge dataset [4] is predominantly comprised of paediatric cases and reveals a noteworthy imbalance for the murmur labels. As shown in Tables 2 and 3, 74% of the patients manifested no heart murmurs, compared to 19% who did. In a minor portion (7%) of the instances, the murmur status remained ambiguous. The outcome label was relatively balanced, with 52% of the samples being labelled normal and 48% being labelled abnormal. As shown in Fig 3, the distributions of age, weight, and height generally conform to expected patterns, with a few outliers (cf. Fig 3).

Table 2 shows a correlation between the occurrence of heart murmurs and abnormal clinical outcomes. However, not all instances of abnormal outcomes can be attributed to heart murmurs, suggesting that other factors also contribute.

### 4.2 Test data analysis

The multi-site evaluation used the PhysioNet 2016 Challenge data as the out-of-distribution data [31, 80]. The dataset is unbalanced; out of the total 3,153 records, only 665 (21%) are classified as abnormal, with the rest (79%) being classified as normal. Gender information is available for 2,689 individuals, 8% of whom are female. Age is present for 2,199 individuals, and ranges from 10 to 90 years, with an average age of 30. However, only 31 records include both height and weight data. Additional demographic information, such as Body Mass Index (BMI), smoking status, and disease severity, are available for distinct subgroups of patients. Data on the location of the recording, the patient's condition, and diagnosed diseases are also occasionally available. As shown in Fig 4, 'Abnormal' recordings are on average significantly ($p < 0.001$) longer (the average length being 25.6 sec) than 'Normal' recordings (the length of which is 21.7 sec).

**Table 2. Murmur labels by outcome labels [n (% of column)].**

|          | Absent      | Unknown    | Present     | Sum         |
|----------|-------------|------------|-------------|-------------|
| Normal   | 432 (62.2)  | 25 (36.8)  | 29 (16.2)   | 486 (51.6)  |
| Abnormal | 263 (37.8)  | 43 (63.2)  | 150 (83.4)  | 456 (48.4)  |
| Sum      | 695         | 68         | 179         | 942         |

**Table 3. Murmur labels by age [n (% of 942)].**

|            | Absent      | Unknown   | Present     | Sum          |
|------------|-------------|-----------|-------------|--------------|
| Neonate    | 4 (0.4)     | 1 (0.1)   | 1 (0.1)     | 6 (0.6)      |
| Infant     | 76 (8.1)    | 25 (2.7)  | 25 (2.7)    | 126 (13.4)   |
| Child      | 495 (52.6)  | 37 (3.9)  | 132 (14.0)  | 664 (70.5)   |
| Adolescent | 53 (5.6)    | 3 (0.3)   | 16 (1.7)    | 72 (7.6)     |
| Missing    | 67 (7.1)    | 2 (0.2)   | 5 (0.5)     | 74 (7.9)     |
| Sum        | 695 (73.8)  | 68 (7.2)  | 179 (19.0)  | 942 (100)    |

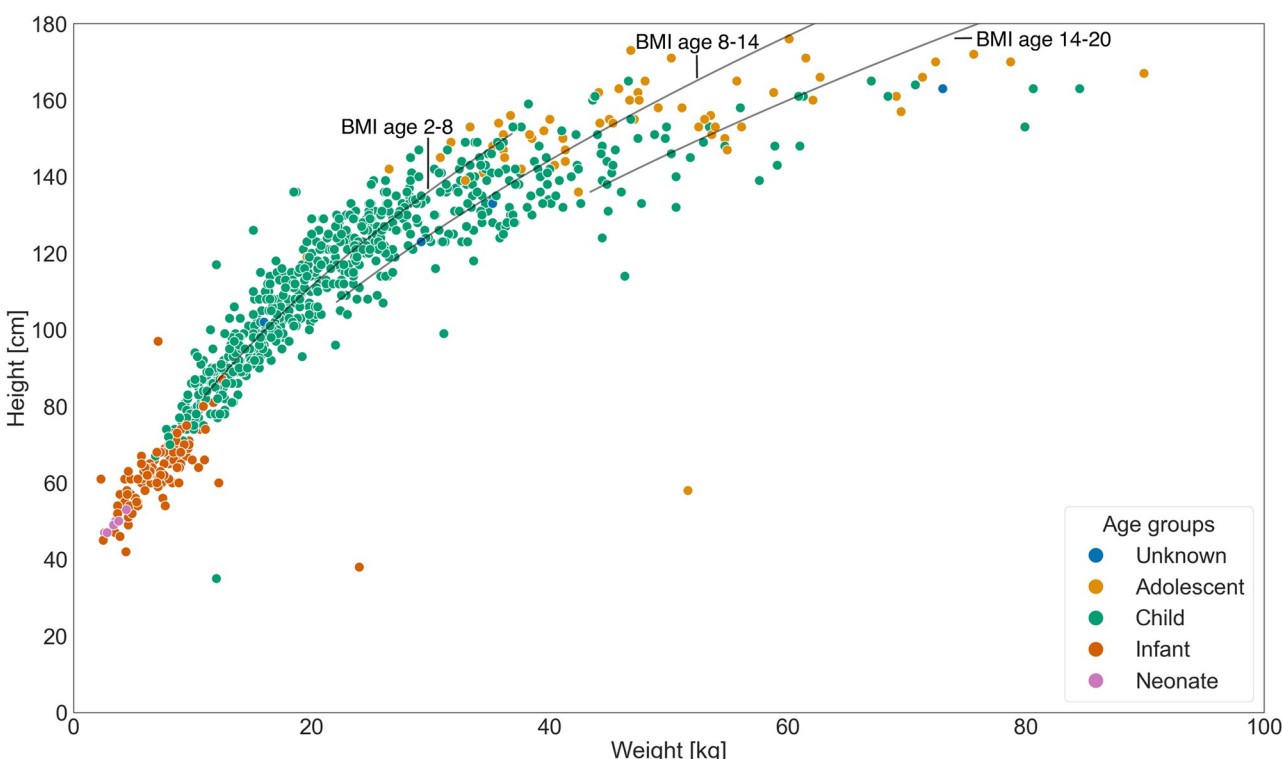

**Fig 3. Distribution of age, weight, and height among patients in the training data (n = 942).** Age groups include Neonate (from birth to 27 days); Infant (from 28 days to 1 year); Child (from 1 to 11 years); and Adolescent (from 12 to 18 years). Black lines indicate height-to-weight combinations corresponding to the medians of the median body mass indices (BMI) as proxy for a healthy BMI, within the 10th to 90th percentile weight range for the three age groups [2, 8), [8, 14), [14, 20]. Data are derived from US sources as cited in Fryar et al. [79], owing to its availability.

$$Height_{[m]} = \sqrt{Weight_{[kg]}/BMI}.$$

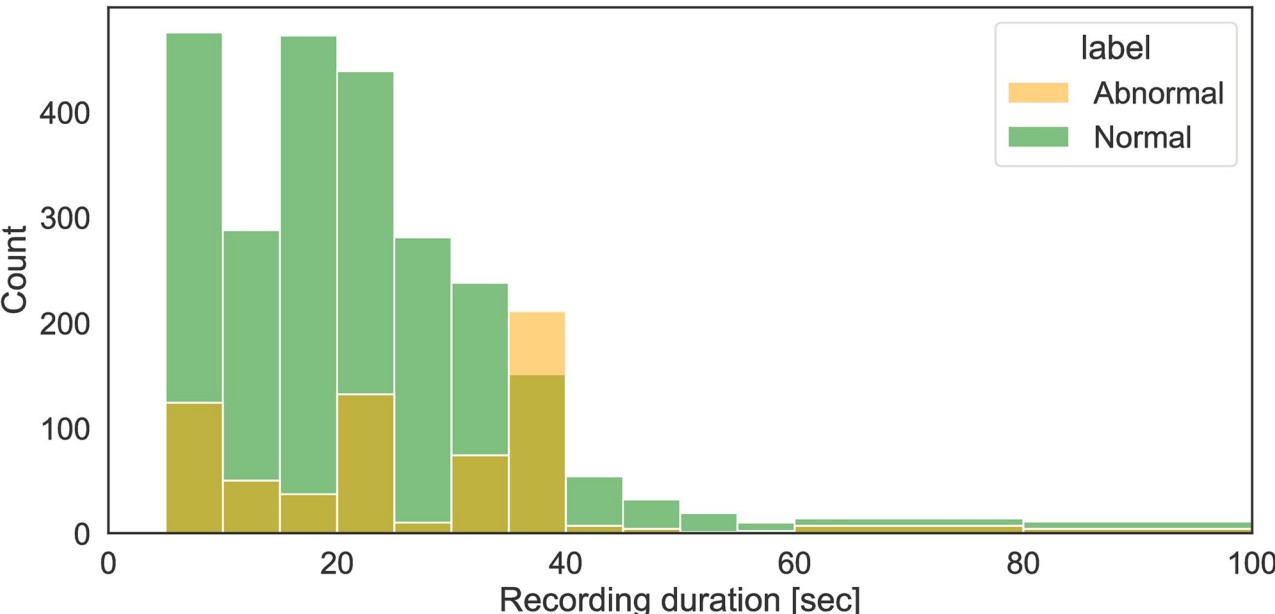

**Fig 4. Distribution of recording lengths by findings in the 2016 Challenge data (Normal: n = 2488, average recording length = 21.7 sec; Abnormal: n = 665, average recording length = 25.6 sec).**

**Table 4. Average performance and standard deviation on ten-fold cross-validation subsets of the training set for various murmur models.** In bold best performing model. BBRes refers to binary Bayesian ResNet as described in Section 3.4. DBRes was the original, multi-class model. Res is the counterpart of BBRes without the Bayesian adjustments.

| Murmur models | Acc. Present/Unknown | Acc. Absent | Overall Accuracy | AUC |
|---|---|---|---|---|
| DBRes binary of multiclass prediction | **0.7030 (0.1821)** | 0.7947 (0.0717) | 0.7770 (0.0743) | 0.8180 (0.1236) |
| Res | 0.4395 (0.1739) | 0.9466 (0.0714) | 0.8151 (0.0749) | 0.8195 (0.1194) |
| Res with dropout during training | 0.4556 (0.2662) | 0.9233 (0.0626) | 0.8197 (0.0332) | 0.8303 (0.0699) |
| BBRes | 0.5033 (0.1823) | **0.9563 (0.0488)** | 0.8408 (0.0614) | 0.8430 (0.1381) |
| BBRes with XGBoost | 0.5185 (0.1914) | 0.9490 (0.0438) | 0.8398 (0.0592) | 0.8379 (0.0540) |
| BBRes with XGBoost, weighted | 0.6268 (0.0838) | 0.9526 (0.0199) | **0.8594 (0.0239)** | **0.8436 (0.0426)** |

**Table 5. Average performance and standard deviation on ten-fold cross-validation subsets of the training set for the outcome model.** BBRes refers to binary Bayesian ResNet as described in Section 3.4.

| Outcome models | Acc. Abnormal | Acc. Normal | Overall Accuracy | AUC |
|---|---|---|---|---|
| BBRes Outcome | 0.4403 (0.1042) | 0.7525 (0.0893) | 0.5976 (0.0653) | 0.6536 (0.1041) |

## 4.3 Model performance analysis

Tables 4 and 5 present an overview of the performance metrics for the various models examined in this paper.

For comparison, the initial model (designed for three-class murmur classification and named DBRes) produced a weighted accuracy of 77.1% (placing it in 4th position) on the hidden test set provided by PhysioNet. It produced only a slightly higher accuracy of 78.0% when evaluated on a locally held-out, stratified subset of the data. The similarity between the murmur challenge scores, obtained from the reserved portion of the training set, and those of the hidden test set from PhysioNet, indicates that the test approach performs consistently with similar screening campaigns.

When the Bayesian approximation model (BBRes) was compared with a pure ResNet model (Res), a clear improvement across all reported metrics was observed (see Table 4). To isolate the effect of adding dropout layers during training from that of retaining them during inference in the Bayesian approach, the results were also compared with a ResNet model where dropout layers were active only during training (Res with dropout). The results indicate that the Bayesian approach still outperforms the approach with only dropouts. (Due to the large standard deviation across splits, none of the differences proved to be statistically significant with a threshold of p<0.01.)

The binary models used consistently demonstrate accuracies and AUC values above 80% for the murmur classification. However, the performance of the outcome model is markedly subpar, with an overall accuracy that fails to surpass 60% (see Table 5). For context, the top ten teams in the Challenge achieved an average accuracy of 56.6% on the hidden test data for the outcome task, with a standard deviation of 1.59%.

As illustrated in Tables 6 and 7, both prediction tasks exhibit substantial error rates, with a pronounced inclination toward false-negative assessments for the presence of abnormalities.

## 4.4 Model generalisability: Multi-site evaluation

Implementing a model in a practical setting requires establishing a decision threshold and formulating rules based on this threshold. As demonstrated in Fig 5, the models exhibit a high

**Table 6. Confusion matrix of the best recordings only model (BBRes), with a decision threshold of 0.5, evaluated on one randomly selected 10% held-out set.** AUC = 0.915, FNR = 0.32.

|  | True present + True unknown | True absent |
|---|---|---|
| Pred. Present/Unknown | 17 | 0 |
| Pred. Absent | 8 | 70 |

**Table 7. Confusion matrix of the best performing, unbalanced model for the outcome label task, using recordings only (BBRes Outcome), with a decision threshold of 0.5, evaluated on one randomly selected 10% held-out set.** AUC = 0.728, FNR = 0.468.

|  | True Abnormal | True Normal |
|---|---|---|
| Pred. Abnormal | 25 | 10 |
| Pred. Normal | 22 | 38 |

sensitivity to decision thresholds. Determining an optimal action point (such as issuing a warning for a follow-up screening) presents a complex challenge. Striking a judicious balance among various types of errors is essential to ensure the model's reliability and effectiveness in real-world applications. (An alternative approach based on ranking all predictions, rather than using a threshold, could entail directing patients with the highest scores to further screenings. However, this strategy would neither be fair to patients nor represent an efficient allocation of resources.)

To investigate the stability of the model on out-of-distribution cases, the model's performance on the outcome task was evaluated with multi-site data from the 2016 PhysioNet

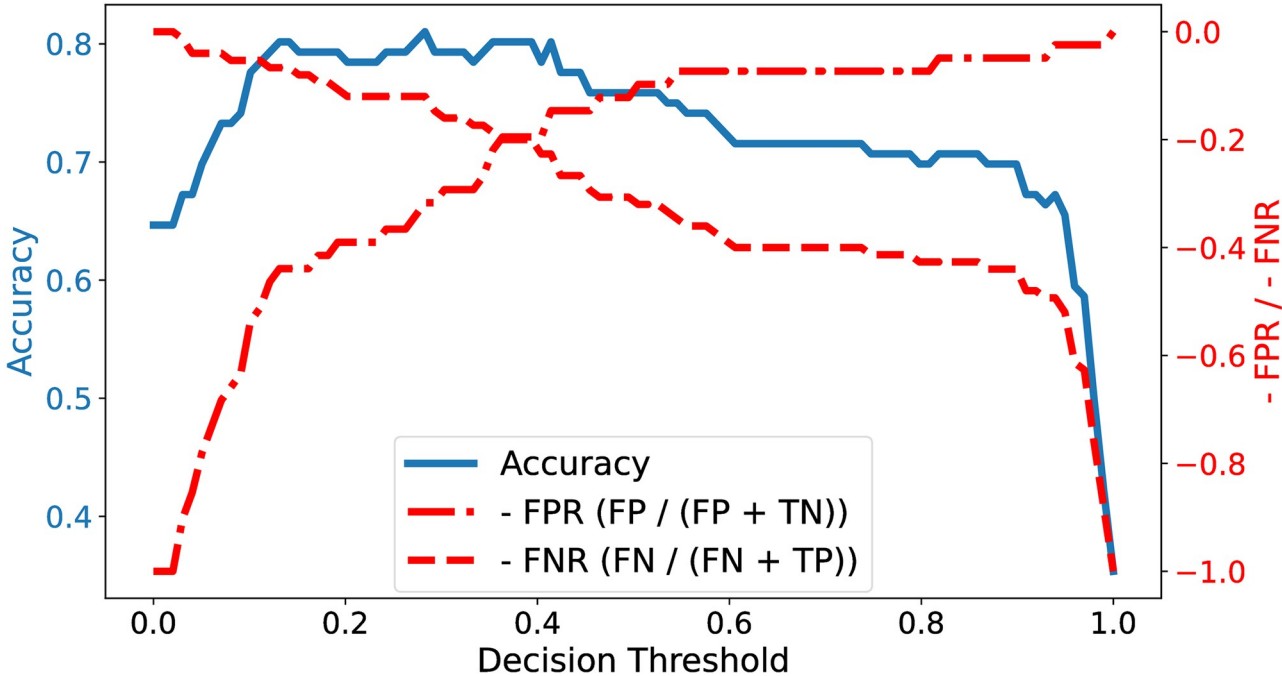

**Fig 5. Accuracy, false-positive-rate (FPR), and false-negative-rate (FNR) of 'Present' label for different decision thresholds of the best performing binary model.** We show the negative FPR and FNR rates such that for both, the rates and the accuracy, the top of the diagram shows the more desired output.

**Table 8. Average accuracy overall, for abnormal class and for normal class of ten-fold cross-validation subsets for the outcome label task, using recordings only, with a decision threshold of 0.5.** We balanced the 2016 training data by sub-sampling.

| Training \Test set | 2016 unbalanced | 2022 |
|---|---|---|
| 2016 balanced | 0.850, 0.749, 0.878 | 0.516, 0.979, 0.062 |
| 2022 | 0.238, 0.778, 0.087 | 0.5976, 0.4403, 0.7525 |

Challenge [31] and the Yaseen dataset [32]. First, we trained the model on the 2022 data and tested it on the 2016 data and vice versa, in a zero-shot fashion. (The 2016 Challenge's leading teams [31] reported accuracies of over 80% with sensitivities and specificities of over 94% and 77% [81].) The findings, displayed in Table 8, show a significant decline in performance when the models were applied to out-of-distribution data, as they almost always assume them to be abnormal. While the results warrant cautious interpretation as the model did not exhibit strong performance on in-distribution data (cf. Table 5) and the data were sourced from different populations, the findings indicate that the deployment of pre-trained models in isolation is unfeasible. To achieve robust performance, one must either standardise the data collection procedure or develop more resilient models, potentially through strategies such as improved feature extraction.

Second, as the Yaseen dataset [32] is often cited for models with accuracy rates exceeding 99%, we also trained and tested our model on the Yaseen dataset, resulting in an accuracy of 99%. Notably, this result was achieved without any hyperparameter tuning or model adjustments; it was simply trained and tested on Yaseen splits. As the Yaseen data are very clean and short, this results shows the strong data dependency of models.

## 4.5 Deployment challenges—NASSS evaluation

By integrating the results of the case study with those of the literature review, the NASSS framework [10] was applied to evaluate the key challenges in deploying AI-supported heart murmur detection in low-income settings. Although a complete evaluation of dimensions 5–7 (organisation, context, and adaptation over time) is not feasible without specific knowledge of the target organisation, Fig 6 indicates that the challenges in the first four dimensions are not highly complex. Regarding the first dimension of NASSS (1A-B), the condition itself (cf. Section 3.5), heart murmurs represent a well-understood medical condition (as described in Section 1) albeit with variations in occurrence, diagnosis, and treatment across different income settings. Concerning the technology (2A-D) and its adoption (4A-C), Oliveira et al. [4] demonstrated that the system can function as a straightforward plug-and-play model requiring minimal staff training. The primary risk identified in this study pertains to the technology's dependability across different sites, operators, and systems. Ensuring standardised data collection through training and quality checks is critical. Furthermore, while the technology is desirable for patients, its financial viability hinges on the specific healthcare organisation within the target country (3A-B).

For comparison, results from other studies are presented (cf. Fig 7). These results must be interpreted with caution due to differences in settings. However, they suggest that although a study may be assessed as relatively straightforward in the initial dimensions, it can encounter complex and complicated challenges related to the adopter and the organisational system.

In summary, the implementation of AI-supported heart murmur detection is feasible under three main conditions: the predictive models must be robust, the organisational

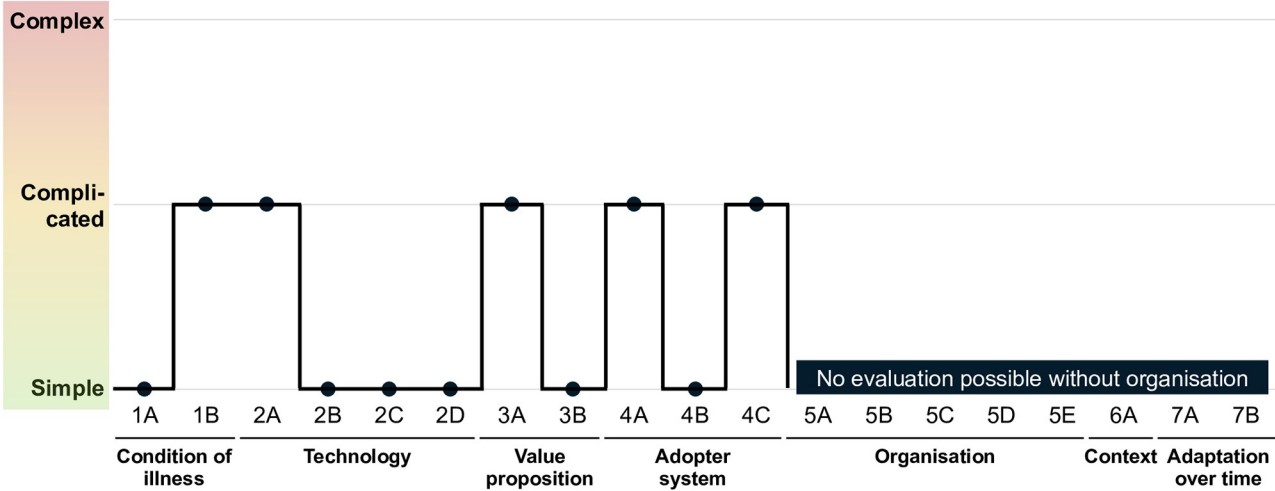

**Fig 6. Indicative NASSS evaluation for the deployment of heart sound recordings in low-resource settings.** (More detailed information is available in S1 Table in the Supporting Information Section.)

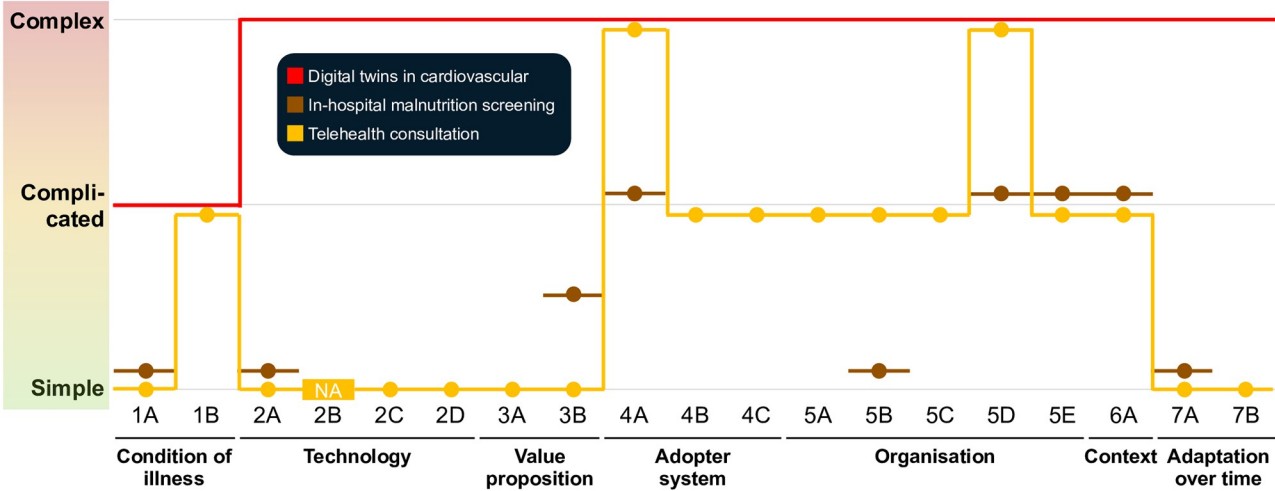

**Fig 7. NASSS evaluation for various examples found in the literature from HIC settings.** Red: Digital twins in cardiovascular medicine (a direct matching to the sub-categories was not possible) [77]; Brown: In-hospital malnutrition screening [78]; Orange: Telehealth consultation [76].

framework must facilitate a sustainable and scalable roll-out (including follow-up care options for patients), and secure funding must be in place.

## 5 Discussion

### 5.1 Cardiac auscultation

Cardiac auscultation offers important opportunities for cardiovascular disease screening. It is a non-invasive, cost-effective, and widely accessible tool that enables primary care physicians to detect abnormal heart sounds (such as murmurs) indicating conditions like valve disorders or hypertrophic cardiomyopathy. This can lead to early intervention and reduced morbidity

and mortality. However, its limitations are notable. Accurate interpretation requires considerable expertise, and even experienced practitioners may struggle to differentiate between benign and pathological murmurs. Auscultation has limited sensitivity and specificity compared to advanced imaging techniques like echocardiography. Background noise, patient body habitus, and variations in heart sounds further complicate its accuracy. Therefore, while valuable for initial screening, auscultation findings often require confirmation with more sophisticated diagnostics for precise diagnosis and treatment.

## 5.2 Automated screening

**Model summary.** In this research, a deep learning technique for identifying murmurs and general irregularities through the analysis of heart sound recordings and demographic information was presented and evaluated. The primary approach, referred to as BBRes (originally named DBRes, or Dual-Bayesian-ResNet, in previous work [7]), employed a binary Bayesian ResNet50 architecture to classify murmurs based on segmented spectrograms of heart audio recordings. This Bayesian model showed marked improvement over a standard ResNet architecture. The extended approach integrates results from BBRes with additional attributes derived from audio signals and patient demographics, using XGBoost for classification. Table 4 demonstrates that spectrograms are an effective data representation and, in combination with ResNet, contribute significantly to predictive performance. The inclusion of demographic information and signal features further improves overall accuracy.

**Opportunities.** The findings of this study underscore the potential of deep neural networks to improve cardiovascular disease research by enhancing the specificity of heart murmur categorisation. This advancement could significantly improve early detection and diagnosis of congenital heart disease, leading to better patient outcomes. Moreover, the development of computational screening methods based on these models promises to streamline diagnostic processes, reduce the need for invasive procedures, and facilitate timely interventions, thereby contributing to more efficient and effective healthcare delivery in cardiology.

**Challenges.** As highlighted in Section 4.3, the performance of our model, as well as that of most challenge models, exhibits low accuracy in the outcome task. This discrepancy is particularly striking when contrasted with results presented in recent literature (cf. Section 1.3). However, the findings from existing studies [8, 9, 82] are not directly comparable to the 2022 challenge. For example, the Yaseen dataset [32] (often cited for models with accuracy rates exceeding 99%) features extremely short (<4 sec) and clean recordings. By contrast, the challenge data include a variety of noises and longer recordings. To investigate these observations further, our model was tested with the Yaseen dataset, resulting in an accuracy of 99%. Notably, this result was achieved without any hyperparameter tuning or model adjustments; it was simply trained and tested on Yaseen splits. In a reverse experiment, the Yaseen model from Nguyen et al. [82] was also tested with the challenge data. The analysis highlighted the challenges in transferring AI models across different datasets in healthcare: although the model from [82] exhibited exceptional performance on the Yaseen dataset, its efficacy significantly diminished when applied to the 2016 challenge data. This observation is critical in understanding the limitations of AI models in healthcare, where data heterogeneity is common.

**Conclusion.** Consequently, the findings indicate the necessity of enhanced focus on data pre-processing, cleaning, robust feature extraction, and standardisation in future research. Incorporating signal quality assessment into the classification pipeline could mitigate these issues. This could prove instrumental in augmenting the cross-site applicability of AI models, ensuring more robust and generalisable healthcare solutions.

### 5.3 Limitations and future research

**Generalisability.** There were two major limitations in this research: A) The models were trained exclusively on children's data, while the out-of-distribution evaluation set predominantly features adult data. B) As demonstrated throughout this paper, developing a model without considering its practical deployment proves unproductive. The choice of the correct loss function for optimisation is highly contingent on the deployment setting, and various loss functions warrant investigation [83]. Nevertheless, the insights gained from this work will likely assist in the identification of areas requiring attention for successful model deployment.

**Feature engineering.** Moreover, this research adopted a methodology focused on directly predicting the target variable using deep learners. This strategy yielded success in the murmur task challenge by emulating the complexities of the weighted, multiclass problem. However, the approach has considerable limitations, notably in model robustness and interpretability. Alternative methodologies based on robust feature engineering, such as segmentation, have been explored by other leading teams [84]. Such robust feature engineering approaches offer potential improvements in interpretability and may enhance model resilience to overfitting [85].

**Multimodality.** Future research could investigate strategies to integrate patient demographic information, signal characteristics, and BBRes outputs more effectively. Additionally, multiple fusion techniques may improve model performance [86]. In this research, features were fused at a relatively late stage. However, an earlier feature fusion could better align with how clinicians use demographic information when interpreting charts [85]. Forthcoming research will consider the types of information that steer the classifier output in multimodal models and will deepen the analysis of unclear cases to determine which cases benefit from specific types of information.

**Model improvements.** Beyond multimodal approaches, there are many other possible paths to explore to increase model robustness. Unsupervised Prediction Alignment (UPA) can assist in maintaining model performance despite variations in data acquisition conditions [87]. UPA uses 'linear piecewise cumulative distribution matching' [87] to align model predictions with reference distributions, ensuring consistent sensitivity and specificity. This approach, adapted from image processing, involves matching cumulative distributions through linear interpolation. It enhances the feasibility of AI model deployment in low-resource settings. The evaluation of foundation models in healthcare applications remains an open area of research [88]. Medical foundation models like BiomedGPT [89] and Med-PaLM M [90] have yet to be tested for tasks similar to ours. Another avenue worth exploring is the replacement of the Fourier method in spectrogram creation with a signature-based approach [91]. The application of self-supervised learning to incorporate more domain-specific data shows promise, as evidenced by a recent paper that introduced HeartBEiT, a vision-based transformer model for ECG analysis [92]. HeartBEiT demonstrated significantly superior performance at lower sample sizes compared to standard CNNs. For an extensive overview of recent developments in heart sound analysis, the work by Ren et al. [93] offers valuable insights.

**Interpretability.** Interpretable machine learning approaches to heart sound classification are a promising and active area of research. One direction involves techniques that highlight the most influential features of heart sound recordings in the decision-making process. Examples include saliency maps [94], SHAP (SHapley Additive exPlanations) values [95], and LIME (Local Interpretable Model-agnostic Explanations) [96]. These techniques have shown promise in rendering AI systems more transparent and understandable, thereby facilitating their adoption in cardiac auscultation and other medical applications. A systematic review by

Ayano et al. (2022) discusses current state-of-the-art research. It highlights the importance of these methods in building trust and providing evidence-based diagnoses [97].

### 5.4 Outlook

For a widespread adoption of automated pre-screening technologies, such as the one studied, several key factors require attention.

One is the implementation of a comprehensive data-mining pipeline (as shown in Fig 8). Such a pipeline typically encompasses several steps: problem comprehension, data

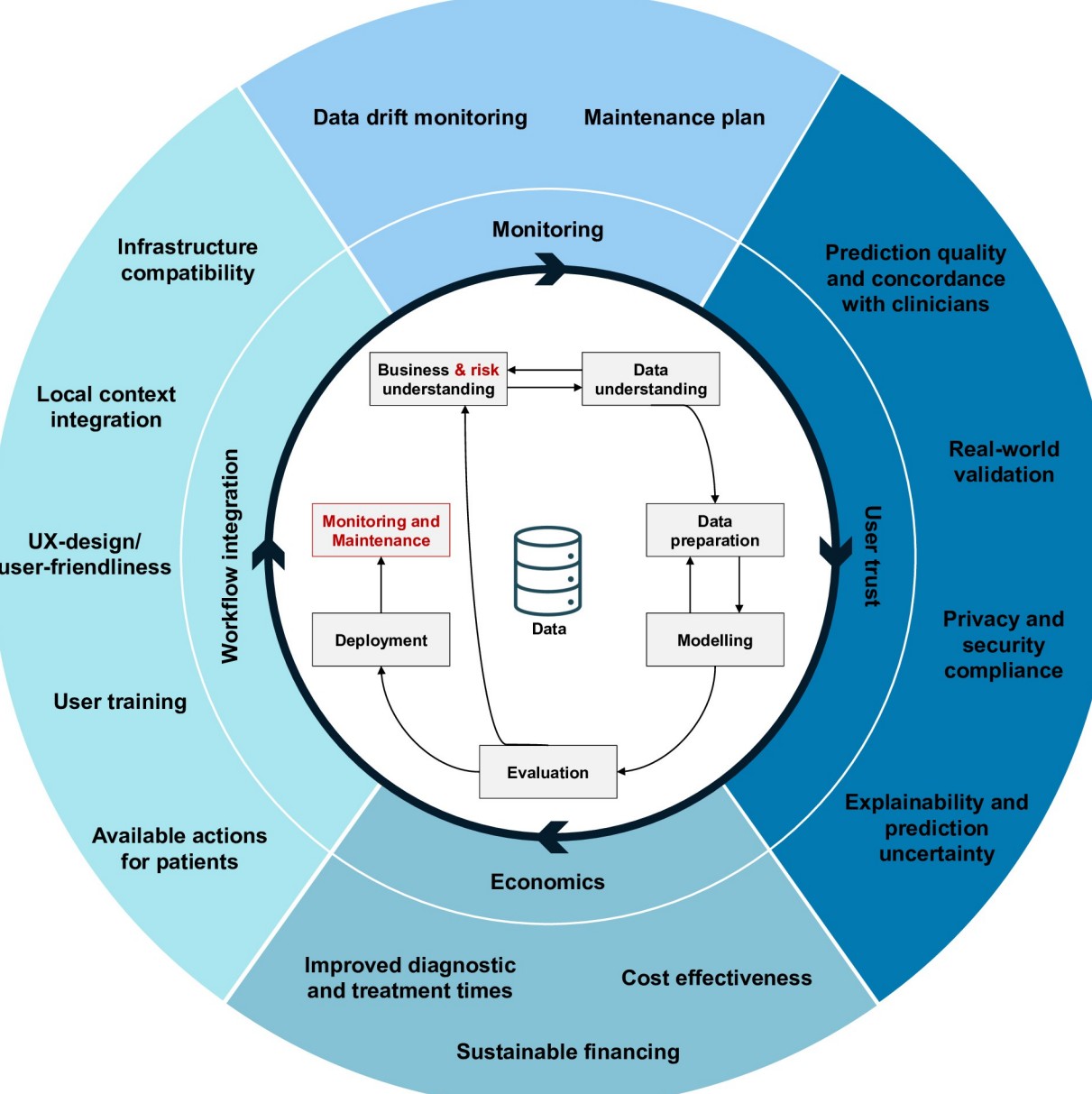

**Fig 8. In blue: Important considerations for a human-centred development.** Centre: Important steps for a successful deployment of AI technologies in healthcare around CRISP-DM, Cross-Industry Standard Process for Data Mining (cf. [98], in red font are author additions to the original process definition).

understanding, data preparation, model training/fitting, evaluation, and deployment (cf. CRISP-DM: [98]). The process should be regarded as cyclical rather than linear to allow for continual refinement. During the initial stage of the problem comprehension, an exhaustive risk assessment proves essential for successful integration of deep learning into sensitive systems. This assessment must encompass the identification of relevant subgroups and potential data correlations. Regular evaluations and monitoring post-deployment contribute to risk minimisation and to successful employment of deep learning applications in sensitive environments.

In addition to the aforementioned pipeline stages, there are several other key aspects to consider:

A). The provision of a user-friendly tool that guarantees reliable data collection and includes a quality check of the data, which is crucial for success (as indicated by Oliveira et al. [4]).

B). The creation of a detailed plan for training operators to collect accurate data and pre-screen patients for eligibility [4, 48]. This plan should specify the methods, timing, and locations for screenings.

C). The assessment of the tool's adaptability to new environments without local fine-tuning, as well as the validation of models on local, representative datasets concerning population and data quality [55, 99]. This should include adaptability to variations in background noise and data collection devices [4].

D). The establishment of a well-defined communication protocol for prediction certainty [17].

E). The implementation of continuous quality monitoring to facilitate timely interventions should performance decline, along with the definition appropriate metrics, which are fundamental for maintaining standards [13, 93].

F). Finally, the introduction of a clear action plan. This ensures that patients understand subsequent steps and that follow-up support is assured.

Regarding point C, recent literature has begun to explore automated correction for variations in data acquisition, such as when different hardware or software are used. Unsupervised alignment methods are some of the proposed solutions to address this issue [87].

By synthesising the literature [16, 17, 35, 93, 100] and the findings presented above, Fig 8 offers an overview of the considerations that are important for deployment. It is crucial that these steps are considered not only during deployment but also throughout the initial problem assessment and the entire process of data collection and modelling.

## Supporting information

**S1 Table. Indicative NASSS evaluation for the deployment of heart sound recordings in low-resource settings.**
(PDF)

## Author Contributions

**Conceptualization:** Felix Krones, Benjamin Walker.

**Data curation:** Felix Krones, Benjamin Walker.

**Formal analysis:** Felix Krones, Benjamin Walker.

**Investigation:** Felix Krones, Benjamin Walker.

**Methodology:** Felix Krones, Benjamin Walker.

**Project administration:** Felix Krones.

**Supervision:** Benjamin Walker.

**Validation:** Benjamin Walker.

**Visualization:** Felix Krones.

**Writing – original draft:** Felix Krones.

**Writing – review & editing:** Felix Krones, Benjamin Walker.

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
