## [Decision Letter · Decision Letter 0]

28 May 2024

PDIG-D-23-00490

From theoretical models to practical deployment: A perspective and case study of opportunities and challenges in AI-driven healthcare research for low-income settings

PLOS Digital Health

Dear Dr. Krones,

Thank you for submitting your manuscript to PLOS Digital Health. After careful consideration, we feel that it has merit but does not fully meet PLOS Digital Health's publication criteria as it currently stands. Therefore, we invite you to submit a revised version of the manuscript that addresses the points raised during the review process.

Please submit your revised manuscript within 60 days Jul 27 2024 11:59PM. If you will need more time than this to complete your revisions, please reply to this message or contact the journal office at digitalhealth@plos.org. Please include the following items when submitting your revised manuscript:

We look forward to receiving your revised manuscript.

Kind regards,

Matthew A Reyna, Ph.D.

Guest Editor

PLOS Digital Health

Journal Requirements:

1. We ask that a manuscript source file is provided at Revision. Please upload your manuscript file as a .doc, .docx, .rtf or .tex.

Additional Editor Comments (if provided):

Reviewers' comments:

Reviewer's Responses to Questions

**Comments to the Author**

1. Does this manuscript meet PLOS Digital Health’s publication criteria? Is the manuscript technically sound, and do the data support the conclusions? The manuscript must describe methodologically and ethically rigorous research with conclusions that are appropriately drawn based on the data presented.

Reviewer #1: Yes

Reviewer #2: Partly

Reviewer #3: Yes

2. Has the statistical analysis been performed appropriately and rigorously?

Reviewer #1: Yes

Reviewer #2: No

Reviewer #3: Yes

3. Have the authors made all data underlying the findings in their manuscript fully available (please refer to the Data Availability Statement at the start of the manuscript PDF file)?

Reviewer #1: Yes

Reviewer #2: Yes

Reviewer #3: Yes

4. Is the manuscript presented in an intelligible fashion and written in standard English?

Reviewer #1: Yes

Reviewer #2: Yes

Reviewer #3: Yes

5. Review Comments to the Author

Reviewer #1: This work considers the general problem of how to leverage AI-based computer-assisted decision (CAD) systems for healthcare in low-income settings, highlighting opportunities and limitations that hinders practical application scenarios.

It also focuses on the specific case of cardiac auscultation POC screening, underlying the impact of lack of generalization of recent machine learning solutions appeared in the literature.

The work is well-written and presents interesting contributions that are currently often overseen in the presentation of AI methods in healthcare, in particular, analyzing the gap between theoretical models and practical applications in low-income settings.

On the other hand, the following issues should be addressed in order to provide a more solid contribution:

 1. While the scope of the analysis of the application of AI-based method for healthcare in low-income countries seems quite broad, the experimental evidence provided in the paper is focused on the specific case of cardiac auscultation. In this sense, I would suggest to narrow the scope of the discussion to this specific case, thus altering slightly the title of the work, and, most importantly, providing a more detailed discussion on the opportunities and limitation associated specifically to cardiac auscultation for cardiovascular disease screening.

 2. When evaluating the generalization capabilities of the different deep learning methods considered, it would be interesting to provide a more systematic set of results, that could involve other public datasets on cardiac auscultation. As a first step, it would be interesting, for example, to repeat the proposed analysis when training on the PhysioNet 2016 dataset and testing on the PhysioNet 2022 dataset, thus providing a more varied training population that do not encompass mostly pediatric subjects.

 3. More details regarding what "effective implementation and conscientious design" (as mentioned in page 2) could be provided in the discussion.

 4. In page 3, the authors mentioned that a binary classification task is considered, motivated by the fact that the proposed method is used as a prescreening tool. However, a significant portion of the heart sound recordings labeled as "unclear" in the PhysioNet 2022 dataset is affected by a severe amount of noise. In this sense, it would be advisable to provide a discussion on the impact of signal quality assessment (as mentioned later in the work) and the possible consequences of treating as abnormal also noisy recordings.

 5. Multimodal data is shown to guarantee improved accuracy and generalizability. However, it would be interesting to showcase more in details what is the impact of the demographic data used for classification, in terms of what kind of information is used to steer the classifier output and which cases are more benefitted by the presence of multimodal data.

 6. In Section 4.5, it is stated that ensuring standardized data collection is critical. In this sense, it would be interesting to reflect on the actual feasibility of this step, especially when considering low-income settings, and to reflect on the interplay between this aspect and unsupervised domain alignment techniques mentioned later in the paper.

 7. It would be interesting to include a discussion about the role of interpretable and explainable models in bridging the gap towards practical applications, overviewing the current state-of-the-art of such solutions for the problem of cardiac auscultation.

Minor issues:

 1. I would suggest to avoid the term POC "monitoring" and use instead POC "screening", as the task considered in the case of cardiac auscultation is that of disease detection and not follow-up.

 2. In page 15, the authors claim that the congruence between the challenge scores and performance obtained in the reserved portion of the training data suggests that their approach effectively generalizes across diverse datasets. I think that this statement should be toned down, as the challenge test data was also collected from the same screening campaigns, in the same conditions as the training set. Therefore, performance on the challenge test set is not representative of performance on diverse datasets.

Reviewer #2: Please find the comments attached as pdf.

The paper, entitled 'From Theoretical Models to Practical Deployment: A Perspective and Case Study of Opportunities and Challenges in AI-driven Healthcare Research for Low-Income Settings' by Felix Heinrich Krones and Benjamin Walker, develops an AI model based on the Binary Bayesian ResNet. This model leverages overlapping log-mel spectrograms of patient heart sound recordings and integrates demographic data. Also, the authors assess the capacity of AI to address healthcare disparities. While the topic of this study is interesting, the paper lacks clarity due to its combination of a review article and a contribution paper, and it does not follow a well-defined narrative path to present its contribution. Unfortunately, I cannot recommend publishing the submitted manuscript in PLOS Digital Health before addressing the following points.

1. In section 1.2, as one of the main contributions of the submitted manuscript authors mentioned “we aim to broaden the discourse by focusing on more fundamental 63 error scores that are straightforward to interpret.” but it is not clear how the presented results can support this aim.

2. In section 2.2, authors about how variation of facility and equipment and data collection can cause different performance of a model in different models. I suggest authors include how different training and basically human accuracy can play a role here. We have inter-rater and intra-rater variability so we basically have an upper bound for how accurate our “ground truth” labels are.

3. The study should include more comprehensive analysis on detecting murmur. In the current format its pretty much the same as “Dual Bayesian ResNet: A Deep Learning Approach to Heart Murmur Detection” published earlier as a proceeding of CinC 2022. I would like to see how the algorithm performs on classifying unclear cases.

4. As one of the main contributions of this paper, the authors mentioned” A case study on developing and evaluating a predictive AI model tailored for low-income environments,” but it's not clear how the developed algorithm contributes to this setting.

5. Authors Allocated a good portion of the manuscript to AI deployment in healthcare (and not related to cardiovascular disease necessarily). I would recommend focusing exclusively on prior research or studies pertaining to cardiovascular disease. Additionally, it should be clarified how the results presented contribute to this research, as it is currently unclear.

Minor comments:

- Figure 1, does not have y axis and it is vague.

- AUC mentioned first in line 138 but the abbreviation was defined in line 341.

- Figure 3 caption, suggest avoiding “=” to show association specially when you used “:” in

Figure 1.

- Figure 4, legend is not clear. I suggest using larger circles and color blind friendly colors.

- Figure 4, add labels for curves/lines in the plot.

- Section names are not well defined.

- Deployment challenges (NASSS). why NASS in parentheses?

- 3.4.2 (Bayesian) Neural Network. why parentheses?

- 4.4 Model robustness . Assessing model robustness -...

Reviewer #3: In this manuscript, the authors conduct (1) a survey of the role of AI methods for addressing healthcare disparities in LMICs and (2) a case study of using AI to screen for heart murmurs from easily-accessible phonocardiograms (PCGs). The case study extends their work in the PhysioNet Challenge 2022, which encouraged teams to develop methods identify heart murmurs in a pediatric population in rural Brazil, and applies the lessons from their case study. This manuscript provides a thoughtful study and application of AI in low-resource environments.

While thoughtful, the manuscript has opportunities for improvement:

1. The survey alternates between a broad survey of using AI to improve healthcare in LMICs and the more targeted application of using AI to screen of PCGs for heart murmurs. Due to the broad scope and difficulty of a comprehensive survey, I recommend conducting a narrower survey and noting when the observations extend more broadly instead of conducting a broad survey and providing narrower examples. In general, the attempt to tie together a broad survey with a specific application is admirable but imperfectly executed.

2. The use of other datasets, i.e., the PhysioNet Challenge 2016 training set, as an out-of-sample holdout set to assess generalizability is helpful. I would recommend switching these datasets as well, i.e., use the PhysioNet Challenge 2016 training set as a training set and the PhysioNet Challenge 2022 training set as a holdout set, to more comprehensively assess the generalizability of this approach to an underrepresented population.

3. Several words, such as "Binary", "Artificial Intelligence", and "Area Under the Curve", are capitalized but probably should not be under the journal's style guide.

4. Some references need additional information to identify the cited resources and for compliance with the journal's style guide.

6. PLOS authors have the option to publish the peer review history of their article (what does this mean?). If published, this will include your full peer review and any attached files.

**Do you want your identity to be public for this peer review?** For information about this choice, including consent withdrawal, please see our Privacy Policy.

Reviewer #1: No

Reviewer #2: No

Reviewer #3: No

---

## [Decision Letter · Decision Letter 1]

17 Oct 2024

From Theoretical Models to Practical Deployment: A Perspective and Case Study of Opportunities and Challenges in AI-driven Cardiac Auscultation Research for Low-Income Settings

PDIG-D-23-00490R1

Dear Mr Krones,

We are pleased to inform you that your manuscript 'From Theoretical Models to Practical Deployment: A Perspective and Case Study of Opportunities and Challenges in AI-driven Cardiac Auscultation Research for Low-Income Settings' has been provisionally accepted for publication in PLOS Digital Health.

Best regards,

Matthew A Reyna, Ph.D.

Guest Editor

PLOS Digital Health

The authors have addressed the issues raised by the reviewers. As Reviewer #1 suggested, I would recommend minor edits to Section 2 to focus more directly on digital auscultation and AI, rather than more broadly AI deployment in healthcare, as suggested during the previous round of reviews, but I do not see a need for an additional round of reviews for these minor edits.

Reviewer Comments (if any, and for reference):

Reviewer's Responses to Questions

**Comments to the Author**

1. If the authors have adequately addressed your comments raised in a previous round of review and you feel that this manuscript is now acceptable for publication, you may indicate that here to bypass the “Comments to the Author” section, enter your conflict of interest statement in the “Confidential to Editor” section, and submit your "Accept" recommendation.

Reviewer #1: All comments have been addressed

Reviewer #2: All comments have been addressed

Reviewer #3: All comments have been addressed

2. Does this manuscript meet PLOS Digital Health’s publication criteria? Is the manuscript technically sound, and do the data support the conclusions? The manuscript must describe methodologically and ethically rigorous research with conclusions that are appropriately drawn based on the data presented.

Reviewer #1: Yes

Reviewer #2: Yes

Reviewer #3: Yes

3. Has the statistical analysis been performed appropriately and rigorously?

Reviewer #1: Yes

Reviewer #2: Yes

Reviewer #3: Yes

4. Have the authors made all data underlying the findings in their manuscript fully available (please refer to the Data Availability Statement at the start of the manuscript PDF file)?

Reviewer #1: Yes

Reviewer #2: Yes

Reviewer #3: Yes

5. Is the manuscript presented in an intelligible fashion and written in standard English?

Reviewer #1: Yes

Reviewer #2: Yes

Reviewer #3: Yes

6. Review Comments to the Author

Reviewer #1: In the revised version of the manuscript, the authors have in general addressed with success the issues raised in the previous round of reviews by this reviewer.

Before considering the document ready for publication, I would suggest to further edit Section 2, in order to provide a more concise review specifically focused on digital auscultation and AI, rather than AI deployment in healthcare, as suggested in the previous round of reviews by several reviewers.

Reviewer #2: The authors have addressed the suggested modifications and have revised the manuscript accordingly.

Reviewer #3: I thank the authors for their responses and changes. The authors have addressed my comments.

7. PLOS authors have the option to publish the peer review history of their article (what does this mean?). If published, this will include your full peer review and any attached files.

**Do you want your identity to be public for this peer review?** For information about this choice, including consent withdrawal, please see our Privacy Policy.

Reviewer #1: No

Reviewer #2: No

Reviewer #3: No
